# Wet carbonate-promoted radical arylation of vinyl pinacolboronates with diaryliodonium salts yields substituted olefins

Chao Wu[1,5], Chongyang Zhao[2,5], Jun Zhou[3,4], Han-Shi Hu[2✉], Jun Li [2], Panpan Wu[3,4] & Chao Chen [1,3,4✉]

Since the landmark work of Heck, Negishi and Suzuki on Pd-catalyzed crossing coupling reactions, innovative discovery of new reactions forming C-C bonds and constructing functional olefins via nonmetal catalysts remains an imperative area in organic chemistry. Herein, we report a transition-metal-free arylation method of vinyl pinacolboronates with diaryliodonium salts to form $C(sp^2)$-$C(sp^2)$ bond and provide trans-arylvinylboronates. The resulting vinylboronates can further react with the remaining aryl iodides (generated from diaryliodonium salts) via Suzuki coupling to afford functional olefins, offering an efficient use of aryliodonium salts. Computational mechanistic studies suggest radical-pair pathway of the diaryliodonium salts promoted by the multi-functional wet carbonate.

[1] Key Laboratory of Bioorganic Phosphorus Chemistry & Chemical Biology (Ministry of Education), Department of Chemistry, and the Graduate School at Shenzhen, Tsinghua University, 100084 Beijing, China. [2] Department of Chemistry & Key Laboratory of Organic Optoelectronics and Molecular Engineering of Ministry of Education, Tsinghua University, 100084 Beijing, China. [3] Environmental Engineering, Wuyi University, Jiangmen 529000, China. [4] International Healthcare Innovation Institute (Jiangmen), Jiangmen 529000, China. [5]These authors contributed equally: Chao Wu, Chongyang Zhao. ✉email: hshu@mail.tsinghua.edu.cn; chenchao01@mails.tsinghua.edu.cn

Vinylboronic esters are highly valuable organic intermediates and are intensively used in various transformations including C–C bond formations[1–3], electrophilic or radical additions, and hydrogenation reactions[4–9]. Among these, the most prominent reaction is Pd-catalyzed Suzuki coupling, which could supply important substituted olefins with aryl, alkenyl, alkynyl, and alkyl halides[2,7,10]. Among the many ways to synthesize multi-substituted olefins (Fig. 1a)[11–14], functional groups are needed to induce the vinyl group of boronates via precedent process or complicated conditions. Among them, the hydroboration of alkynes has gained much attention owing to efficiently access to arylvinylboronates via employing transition metal such as copper[15,16], silver[17], ruthenium[18], etc. as catalyst (Fig. 1b-a). In addition, metal-photocatalyzed borylation reaction of vinyl halides has also been developed in recent years (Fig. 1b-b)[19,20]. Apparently, the direct modification of C–H on vinyl group is the most attractive way due to the efficiency. However, there is a big challenge for this strategy since the coupling reactions of aryl-electrophile with vinylic C–H bonds are normally catalyzed by Pd-catalyst (Heck-type reaction) (Fig. 1b-c)[21,22], under which reaction conditions, boronate groups are generally not tolerant and thus such transformation is hardly realized[23–26]. Herein we report a wet base-promoted reaction of vinyl pinacolboronates and diaryliodonium salts (Ar[1]I[+]Ar[2]OTf[−]) to afford the corresponding trans-arylvinylboronates with high yields and selectivity. Our new findings disclose the radical arylation of vinyl pinacolborate 2 can be realized with diaryliodonium salts (Ar[1]I[+]Ar[2]OTf[−]) 1 promoted by wet base (such as carbonate, typically), so it is characterized by the simplicity and the possibility of further functionalization (Fig. 1c). Consequently, a new pathway for efficient employment of both aromatic moieties of (Ar[1]I[+]Ar[2]OTf[−]) is realized, engaged in two types of C–C bond forming reactions in the iterative synthesis of olefins[27–29].

Recently, diaryliodonium salts, $Ar_2I^+X^-$, have received considerable attention due to their powerful arylation for various nucleophiles to synthesize valuable aromatic compounds. A big challenge for these arylation reactions is how to efficiently use both aryl groups of the diaryliodonium salts since only one aryl group was utilized and the other one was deposited in most cases[29]. On the basis of our group's previous research on diaryliodonium salts[30–34], here we report a tandom process in which an aryl iodide generated in situ is captured in a second step by a Suzuki reaction, yielding aryl olefins. This atom-economical use of diaryliodonium salts may offer a useful approach to the

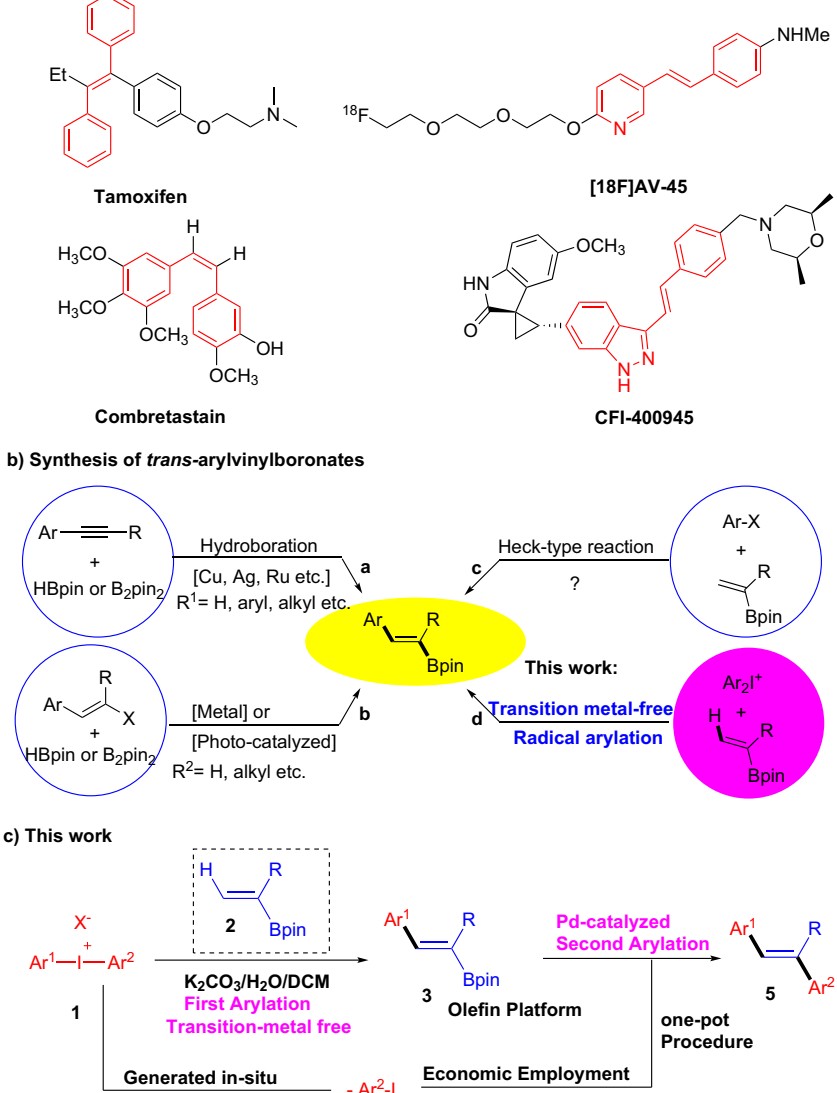

**a) Representative bioactive derivatives containing arylvinyl groups.**

**Tamoxifen**

**[18F]AV-45**

**Combretastain**

**CFI-400945**

**b) Synthesis of trans-arylvinylboronates**

This work:
**Transition metal-free**
**Radical arylation**

**c) This work**

**First Arylation**
**Transition-metal free**

K₂CO₃/H₂O/DCM

**Olefin Platform**

**Pd-catalyzed**
**Second Arylation**

**Economic Employment**

**one-pot Procedure**

Generated in-situ

**Fig. 1 Arylvinylboronates and olefins. a** Multi-substituted olefins. **b** Synthesis of trans-arylvinylboronates. **c** This work for a novel iterative synthesis of olefins.

**Table 1 Optimization of the reaction condition for the arylation of vinylboronic esters.[a]**

| Entry | Catalyst | Base (1 eq.) | Additive | Solvent | Temp. (°C) | Yield[b] (%) | |
|---|---|---|---|---|---|---|---|
| | | | | | | 3a | 4a |
| 1 | CuCl | DIPEA | – | DCE | 80 | Trace | <1 |
| 2 | CuCl | $K_2CO_3$ | – | DCE | 80 | Trace | 8 |
| 3 | CuCl | $K_2CO_3$ | TBAF[c] | DCE | 80 | 3 | 2 |
| 4 | CuCl | $K_2CO_3$ | $H_2O$ | DCE | 80 | 62[d] | 4 |
| 5 | $Pd(OAc)_2$ | $K_2CO_3$ | $H_2O$ | DCE | 80 | 11 | 58 |
| 6 | $Pd(PPh_3)_4$ | $K_2CO_3$ | $H_2O$ | DCE | 80 | 8 | 67 |
| 7 | – | $K_2CO_3$ | $H_2O$ | DCE | 80 | 64 | <1 |
| 8 | – | $K_2CO_3$ | $H_2O$ | DCM | 80 | 81 | <1 |
| 9 | – | $K_2CO_3$ | $H_2O$ | PhMe | 80 | 61 | <1 |
| 10 | – | $K_2CO_3$ | $H_2O$ | DCM | 90 | 84 | <1 |
| **11** | **–** | **$K_2CO_3$** | **$H_2O$** | **DCM** | **100** | **89(88)[e]** | **<1** |
| 12 | – | $K_2CO_3$ | $H_2O$ | $CH_3OH$ | 100 | Trace | 0 |
| 13 | – | $K_2CO_3$ | $H_2O$ | THF | 100 | Trace | 0 |
| 14 | – | $K_2CO_3$ | $H_2O$ | DMF | 100 | Trace | 0 |
| 15 | – | – | $H_2O$ | DCM | 100 | NP[f] | NP[f] |
| 16 | – | $Li_2CO_3$ | $H_2O$ | DCM | 100 | 84(82)[e] | <1 |
| 17 | – | $Ag_2CO_3$ | $H_2O$ | DCM | 100 | NP[f] | <1 |
| 18 | – | $KH_2PO_4$ | $H_2O$ | DCM | 100 | Trace | <1 |

[a]Unless noted, reactions were performed with **1a** (0.15 mmol), **2a** (2.0 eq.), catalyst (10 mol%), and additive (40 eq.), in solvent (1 mL) at the temperature described.
[b]Determined by GC analysis using n-dodecane as an internal standard.
[c]2.0 equiv. of TBAF was used.
[d]E/Z isomer of **3a** was 2/1.
[e]Isolated yield.
[f]No product.
The optimal conditions are in bold.

iterative synthesis of aryl olefins using alkenyl boronic esters as intermediates (Fig. 1c). Of note, it will be a novel approach for the iterative synthesis of aryl olefins using alkenyl boronic esters as intermediate.

## Results and discussion

**Investigation of reaction conditions**. To achieve this goal, we initially examined the reaction of di(4-tolyl) iodonium triflate **1a** and pinacol vinylboronate **2a** serving as model substrates. As shown in Table 1, no product or low yield was observed when the reaction was performed at 80 °C in DCE with CuCl as catalyst and 1.0 equivalents of DIPEA or potassium carbonates as base. The desired product **3a** could be detected in 3% yield (determined by GC with n-dodecane as internal standard) in the presence of additive of tetra-butylammonium fluoride (TBAF). It was pleasingly found that the yield could be improved when the 40 equivalents of water was employed as the additive, affording **3a** in 62% yield (in a mixture of Z− and E− isomers, Table 1, entry 4). As a comparison, Pd-catalyzed systems tended to give the Suzuki-type coupling product **4a** (Table 1, entries 5 and 6). In addition, it was surprised to find that the reaction could also work without CuCl catalyst to give the only E-isomers in 64% yield (entry 7), indicating that the method was a novel approach for constructing $C(sp^2)$–$C(sp^2)$ bonds. The element analysis showed that the content of transition metal was below 5 ppm. Subsequently, various solvents including DCM, PhMe, THF, $CH_3OH$, DMF (entries 8-9, 12-14) were screened. As a result, DCM was the best choice, affording **3a** in 81% yield (entry 8). The yield was increased when elevating the reaction temperature, eventually, affording the isolated yield in 88% at 100 °C in a sealed tube (entry 11). Further base screening of inorganic base such as $K_3PO_4$, $NaHCO_3$, and $Li_2CO_3$ proceeded compatibly

(Supplementary Table 1), and $Li_2CO_3$ also gave the isolated yield of 82%. Of note, treatment of the reaction using insoluble $Ag_2CO_3$ as base did not give the desired product **3a** (entry 17), and the control experiment revealed that no reaction occurred in the absence of base (entry 15). Moreover, increasing or decreasing the equivalent of water significantly resulted in reduced yields. These results proved that water could dramatically influence the reaction. In addition, the counter anion of $Ar^1I^+Ar^2OTf^-$ **1** including $^-OTs$ and $^-OAc$ could give comparable yield of **3a** (Supplementary Table 1).

**Substrate scope**. Under the optimized condition, the scope of this novel procedure was sought to be investigated. Initially, functionalized diaryliodonium salts **1** with a broad range of substitutions were examined. As shown in Fig. 2, substitutions with diverse functional groups such as (o, m, p−) methyl (**1a,1k,1m**), halogen (**1b–1d,1j,1l**), tert-butyl (**1g**), trifluoromethyl (**1i**), and methoxy (**1h**) were all well-tolerated, affording the corresponding products with good stereoselectivity in moderate to good yields. Notably, the substrates **1n** and **1o**, bearing a steric hindrance substituents, were also furnished in 82% (**3n**) and 55% (**3o**) yields. Interestingly, unsymmetrical diaryliodonium salts ($ArI^+MesOTf^-$) containing 2-chloropyridine (**1q**), 2-naphthalene (**1p**), and 4-biphenyl(**1r**) species also gave corresponding products rather than product **3o**, which might account for that steric effect was more obvious than electronic effect in terms of the two competing aryl motifs. Next, a series of substituted vinyl pinacolboronates were reacted with **1a**. It was delighted to find that simple alkyl substitutions such as methyl(**2t**), ethyl(**2u**), propyl(**2v**) all went with moderate to good yields, while 2-phenyl substitution only afforded the product **3w** in 27% isolated yield. It was worth mentioning that (Z)-formation of 1,2-substuited alky olefin **2x**

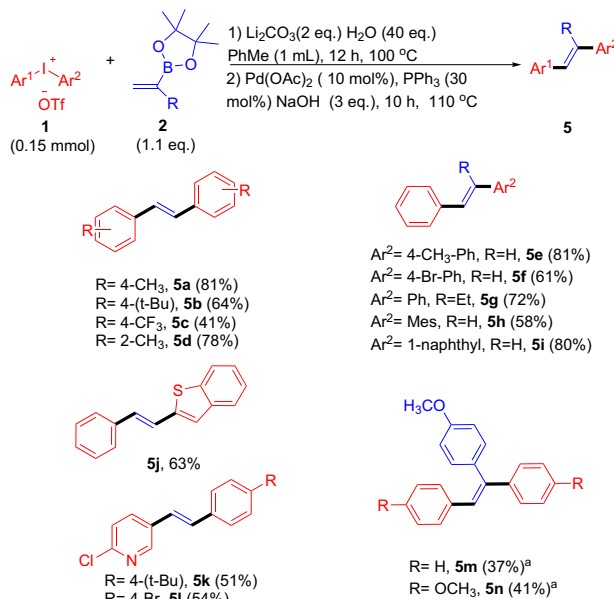

**Fig. 2 The scope of diaryliodonium salts and pinacol vinylboronate.** Reactions were performed with **1** (0.15 mmol), **2** (2.0 eq.), base (1.0 or 2.0 eq.), and $H_2O$ (40 eq.), in DCM (1 mL) at the temperature 100 °C.

**Fig. 3 The synthesis of various di- or tri-substituted aryl olefins.** [a]1.2 eq. of **2**.

could be productive to exclusively give the product **3x** in 41% yield, whereas no product was afforded while (E)-formation of olefins *trans*-**2x** served as substrates.

After the successful arylation of C–H bond of vinyl pinacolboronates **2** was realized with $(Ar^1I^+Ar^2OTf^-)$ **1**

promoted by wet base, we were keen to explore the further arylation of products **3** for efficient employment of both aromatic moieties of **1** via Suzuki reactions. Thus, a series of palladium catalyst, the temperature and phosphorus ligands were screened (Supplementary Table 2), and the best isolated yield and superior selectivity was obtained with $Pd(OAc)_2$ as the catalyst, NaOH as base, $PPh_3$ as ligand, of which **5a** was given in 81% isolated yield (Fig. 3)[2,35]. Then, this one-pot protocol was extended to other substrates. As desired, symmetrical diaryliodonium salts with a variety of substituents (o,p-methyl, p-tert-butyl, p-trifluoro-methyl) were all accomplished smoothly to give products **5a**–**5d**. Subsequently, various unsymmetrical substituted diaryliodonium salts and substituted alkenyl borate esters were examined. It was all productive for various diaryliodonium salts when either the methyl (**5e**), bromo substituents (**5f**) or a bulky aryl group such as 2,4,6-trimethyl (**5h**) or naphthalene (**5i**) moiety. In addition, hetero-aromatic rings including benzothiophene (**5j**) and pyridine species (**5k**, **5l**) were well-tolerated to offer the respective products. Tri-substituted olefins were also afforded with comparable yields (**5g**, **5m**, **5n**), which were ubiquitous building blocks (vide infra).

**Mechanistic study**. To investigate the mechanism of the arylation on C–H bond of **2** with **1**, a few experiments were conducted. First, the effect of $K_2CO_3$ amount was investigated in this process. The control experiments (Supplementary Figs. 1 and 2) showed that there was a dramatic rate increase after 1 to 2 h when 1 equivalent of $K_2CO_3$ base was used and reached >60% yield; as a comparison, increasing the amounts of $K_2CO_3$ led to an evident rate decrease. Due to less solubility, 2 equivalent of $Li_2CO_3$ was

**Fig. 4 Preliminary mechanistic study. a** Radical trapping experiments. **b** Radical detected experiments by EPR. **c** Proposed mechanism for arylation of vinylboronic esters.

needed. Above results indicated that the base amount was crucial to this procedure that might be essential for activating alkenyl borates **2** and accelerating dissociation of the OTf group of **1**[36–39]. To elucidate this transformation, 2,2,6,6-tetramethyl-1-piperidinyloxy was introduced to this base-promoted aryl migration process, and the adduct 2,2,6,6-tetramethyl-1-phenylpiperidine **6** was detected[40,41], and it could be even obtained in higher yields in the absence of **2a** (Fig. 4a). Moreover, the deficient of either base or water could not be capable of getting the desired product **3a** and the trapping product **6**. The above results were consistent with the EPR experiments (Fig. 4b and Supplementary Fig. 3), which indicated that the base-$H_2O$ system could release $CO_3^{2-}$ and split diaryliodonium salts into a pair of carbonate-stabilized radical **7a** (vide infra) and aryl radicals **8a**[40,42–44].

On the basis of above results, a plausible mechanistic pathway was proposed in Fig. 4c. $Ph_2I^+OTf^-$ **1e** was triggered by $CO_3^{2-}$ to give **I**, which accelerated radicals **8e** and $PhI-CO_3^-$ formed via homo cleavage. The intermediate **II** formed from vinylboronate **2a** and $CO_3^{2-}$ enabled the addition of radicals **8e** to form the C–C bond[45–47], generating the "ate" α-boronate adduct radical species **III**[48–52]. Species **III** was capable of occurring SET reaction with radical $CO_3^-$ to give the intermediate moiety **IV** and intramolecular dehydrocarbonate give the "ate" intermediate **VI**, which further eliminated to give the desired product **3e**.

To gain further insight of the base-promoted pathway, density functional theory (DFT) calculations were carried out to explore the reaction mechanism. In this system, the role of $H_2O$ in the reaction was discussed in Supplementary Figs. 4 and 5 and

Supplementary Note 1. Besides, the anion exchange of diaryliodonium salts **3e** from $OTf^-$ to $CO_3^{2-}$ is easy to generate the intermediate **I** with quite an exothermic reaction energy release of 39.7 kcal/mol (Supplementary Fig. 6). The complex $Ph_2I^+X^-$ (X = $K_2CO_3$, $KCO_3^-$, $CO_3^{2-}$, $OTf^-$) decompose to $Ph-I^+X^-$ radical and phenyl radical $Ph^•$ endothermically. The corresponding Gibbs free energy required for the decomposition follows the order: $K_2CO_3 > OTf^- > KCO_3^- > CO_3^{2-}$ (Supplementary Figs. 7 and 8). The Gibbs free energies for the case X = $K_2CO_3$ and $OTf^-$ are +90.7 and +32.6 kcal/mol, respectively, which are so high that the decomposition can hardly take place under the experimental condition. While on the other hand, the Gibbs free energy of +2.5 kcal/mol for the case X = $CO_3^{2-}$ coming from the ionization of $K_2CO_3$ by water is small enough for the subsequent homo cleavage, the fact that no reaction occurs without water addition to the system. The proposed mechanism suggests that it starts from the combination of vinyl pinacolboronates and carbonate to form the intermediate **II**, which is calculated to be exothermic by 7.4 kcal/mol. The phenyl radical attacks the =$CH_2$ group of **II** to give the intermediate **III**, which is calculated to be exothermic by 12.4 kcal/mol (Supplementary Figs. 9 and 10).

The remaining calculated pathway for the reaction is shown in Fig. 5 and Supplementary Data 1, which shows that the intermediate **III** reacts with $PhI-CO_3^-$ radical to produce the intermediate **IV**. Due to the weak interaction of PhI with $CO_3^{•-}$ radical anion, PhI will directly leave the reaction system and $CO_3^{•-}$ is bonding to intermediate **III** simultaneously. The next step of the reaction is the rate-determining one with a barrier of 25.5 kcal/mol

**Fig. 5 The calculated Gibbs free energy profile.** The reaction pathway of the arylation of vinyl pinacolboronates.

**Fig. 6 Chemical derivatives.** The synthetic efforts toward: **a** [18F] AV-45; **b** Chlorotrianisene; **c** tamoxifen.

overcoming the **TS1**, corresponding to proton abstraction by the CO₃ group and giving the intermediate **V**. The O–H bond length 1.06 Å of the intermediate **V** is longer than that of bicarbonate anion (0.97 Å), indicating that the benzylic proton is not abstracted completely, as long as the basicity of carbonate is not large enough. Then, **V** could convert into **VI** through the **TS2**. In this process, with the CO₃ group bonding to the α carbon left, the charge transfer occurs from the benzylic carbanion to the leaving CO₃ group, leading to the formation of C=C bond by overcoming a very small barrier of 6.2 kcal/mol. In the **TS2**, the distance between the leaving CO₃ group and α carbon is 2.43 Å, and the formed C=C bond length is 1.39 Å, which is approximately equal to that of $C_2H_4$ (1.33 Å). Subsequently the product **3e** is obtained from **VI** via releasing bicarbonate, bearing successive barriers of 5.0 kcal/mol. The DFT pathway shows that the relative location $H_a$ and $H_b$ of intermediate **IV** can be attributed to the stereo-configuration of the final product, because the carbonate bonded to the boron atom abstracts the $H_a$ atom, while the $H_b$ atom remains. The $H_b$ and $H_c$ atoms are in the opposite direction along the C–C bond. Besides, we have also considered the situation that reaction starts from the binding of **2a** and **$KCO_3^-$** (the detail in Supplementary Fig. 11), which

shows a less preferred reaction mechanism comparing with that of **$CO_3^{2-}$**.

The skeletons of aryl olefins widely occur in many biologically active compounds. The potential utility of this method was also assessed, as shown in Fig. 6, some illustrative cases were accomplished. The product **5l** could be precisely transformed into [18F]AV-45, an effective PET agent for targeting Aβ plaques in human cerebrovascular, under the standard conditions of the Fig. 3[53]. In addition, Chlorotrianisene **10** was furnished from compound **5n** with 98% yield in one step of chlorination reaction[54]. Finally, the one-pot process of constructing tri-substituted olefins was applied to the synthesis of (Z)-tamoxifen precursor **5m** with good selectivity, and then a series of downstream reactions were manipulated to afford the (Z)-tamoxifen in 68% yield[55,56].

In summary, we have developed an approach for selective arylation of C–H bond of vinyl pinacolboronates utilizing diaryliodonium salts and water-base as additive. This new strategy was exemplified of two-component arylation of diary-liodonium salts accessing aryl olefins via radical-type and Suzuki-type cross-coupling reaction in one-pot, which has been demonstrated as an iterative synthesis of multi-substituted

olefins. The mechanistic experiments and DFT theoretical studies revealed the multi-function of carbonate and a novel radical-pair pathway of diaryliodonium salts promoted by wet carbonate.

## Methods

**General considerations**. Unless specified, all substrates were obtained commercially from various chemical companies and their purity has been checked before use. Unless otherwise stated, all commercial reagents were used as received without purification. The synthesis of **3**: mixture of diaryliodonium salt (0.15 mmol, 1.0 eq.) and base [condition a: $K_2CO_3$ (1 eq.) condition b: $Li_2CO_3$ (2 eq.)] was added into a schlenk tube and then evacuated and recharged with $N_2$ for three times. After that, 1.0 ml DCM was added in, followed by vinyl pinacol boronic esters (0.30 mmol, 51 μl) and pure water (6.0 mmol, 100 μl). The tube and mixture were stirred at 100 °C for 12 h. After completion, the tube was cooled to room temperature, then NaCl aq. (10 ml) was added and the mixture was extracted with EtOAc (10 ml × 3), and then dried by anhydrous $Na_2SO_4$. The mixture was evaporated then purified on silica gel (petroleum ether/EtOAc = 50:1) provided the corresponding product. Full experimental details can be found in the Supplementary Methods. NMR spectra can be found in Supplementary Figs. 12–50.

## Data availability

The authors declare that the data supporting the findings of this study are available within the article and its Supplementary Information and Supplementary Data 1 files. All relevant data are also available from the authors.

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

## Acknowledgements

This work was supported by the National Key Research and Development Program of China (2016YFB0401400), the National Natural Science Foundation of China (21871158, 91645203, and 21672120), the Fok Ying Tong Education Foundation of China (Grant No. 151014), the Department of Education of Guangdong Province (No. 2016KCXTD005), and the Youth Foundation of Wuyi University (No. 2017td01).

## Author contributions

C.W. discovered the reaction and performed the optimization. C.W., J.Z., and P.W. investigated the scope of the substrate and performed the application. And C.Z., H.S.H. and J.L. carried out computational studies. C.C. designed and directed the whole project and wrote the paper with input from all authors. All authors analyzed the results and commented on the paper.

## Competing interests

The authors declare no competing interests.
