## [Peer Review File · Communications Chemistry]

Reviewers' comments:

Reviewer #1 (Remarks to the Author):

This manuscript describes carbonate-promoted radical arylation of vinyl pinacolboronates with diaryliodonium salts to synthesize substituted olefins. It's very meaningful to demonstrate transition-metal-free arylation method of vinyl pinacolboronates, which could further proceed Suzuki coupling to afford functional olefins. Therefore, it is worthy of publication. Nevertheless, there are some critical issues to be addressed.

1. Please explain the role of H₂O in the reaction. Does H₂O just increase the solubility of base, or does it have other effects to promote the reaction?
2. Would a series of mixture be obtained when unsymmetrical substituted diaryliodonium salts and substituted alkenyl borate esters were examined in Figure 3? Could two different aryl radical be detected if unsymmetrical substituted diaryliodonium salts was examined like Figure 4Ab?

It might be possible to synthesize trisubstituted aryl olefins by control electrical properties of substituted diaryliodonium salts.

Reviewer #2 (Remarks to the Author):

Functionalization of olefins via nonmetal catalysts remains challenge in organic chemistry. In this work, a base assisted transition-metal-free arylation method was reported by the author. This strategy enable vinyl pinacolboronates transformed in one pot procedure to give the desired diarylation olefins with diaryliodonium salts. Furthermore, a novel base assisted radical-pair pathway was also proposed in this paper. Whilst the claim that the authors make is interesting to the community, I don't feel they've done enough to support and expand on it. A detailed reaction mechanism for the formation of carbonate-stabilized radical 7a and followed radical addition steps should also given by the DFT calculation as they proposed in figure 4B. I also don't distrust their energy results, however their discussion of the results could be improved. On the whole, I think this paper is suitable for publication on Communications Chemistry with minor revision.

The followings are several questions and suggestions

1. In Figure 2, the amount of base is different when K₂CO₃ and Li₂CO₃ were used respectively. Can author give a pellucid explanation on this phenomenon. May be the effect of Lewis acid K⁺ or Li⁺ should also be considered in the mechanism. Although the author have discussed the decomposition processes with K⁺ in supporting information, the reaction model for this special step is not enough. The coordination number of K⁺ is unsaturated and the single K⁺ model should also be taken into account in the calculation (J. Am. Chem. Soc. 2017, 139, 29, 9909-9920).
2. In figure 5, for the deprotonation step, the current transformation from intermediate IV seems unreasonable to give a relatively unstable intermediate V, May be the outer sphere deprotonation pathway is favorable than the current inner sphere pathway. Besides, another carbonate on the C atom can also be used for the deprotonation step. Furthermore, the detailed mechanism for the generation of radical 7a and aryl radical must be given in the PES.

Comments:

Functionalization of olefins via nonmetal catalysts remains challenge in organic chemistry. In this work, a base assisted transition-metal-free arylation method was reported by the author. This strategy enable vinyl pinacolboronates transformed in one pot procedure to give the desired diarylation olefins with diaryliodonium salts. Furthermore, a novel base assisted radical-pair pathway was also proposed in this paper.

Whilst the claim that the authors make is interesting to the community, I don't feel they've done enough to support and expand on it. A detailed reaction mechanism for the formation of carbonate-stabilized radical **7a** and followed radical addition steps should also given by the DFT calculation as they proposed in figure 4B. I also don't distrust their energy results, however their discussion of the results could be improved. On the whole, I think this paper is suitable for publication on *Communications Chemistry* with minor revision.

The followings are several questions and suggestions

1. In Figure 2, the amount of base is different when K_2CO_3 and Li_2CO_3 were used respectively. Can author give a pellucid explanation on this phenomenon. May be the effect of Lewis acid K^+ or Li^+ should also be considered in the mechanism. Although the author have discussed the decomposition processes with K^+ in supporting information, the reaction model for this special step is not enough. The coordination number of K^+ is unsaturated and the single K^+ model should also be taken into account in the calculation (J. Am. Chem. Soc. 2017, 139, 29, 9909-9920).

2. In figure 5, for the deprotonation step, the current transformation from intermediate **IV** seems unreasonable to give a relatively unstable intermediate **V**, May be the outer sphere deprotonation pathway is favorable than the current inner sphere pathway. Besides, another carbonate on the C atom can also be used for the deprotonation step.

Furthermore, the detailed mechanism for the generation of radical **7a** and aryl radical must be given in the PES.

Reviewer #1 (Remarks to the Author):

This manuscript describes carbonate-promoted radical arylation of vinyl pinacolboronates with diaryliodonium salts to synthesize substituted olefins. It's very meaningful to demonstrate transition-metal-free arylation method of vinyl pinacolboronates, which could further proceed Suzuki coupling to afford functional olefins. Therefore, it is worthy of publication. Nevertheless, there are some critical issues to be addressed.

1. Please explain the role of H₂O in the reaction. Does H₂O just increase the solubility of base, or does it have other effects to promote the reaction?

Response: Thank you for your helpful suggestions, following which we have added more calculations by adding one H₂O molecule into complex IV and the results are shown in Figure 1 below. It shows that the addition of one H₂O molecule step is endothermic by 13.7 kcal/mol. The following step of proton transfer (PT) requires further 27.0 kcal/mol free energy barrier. This H₂O molecule plays a role in the proton transfer between benzyl site and carbonate bonded to the boron atom. The total barrier including binding energy of the water-catalyzed proton abstraction is high as to 40.7 kcal/mol. Therefore, we think the H₂O molecule does help to activate the reactions.

Figure 1. The potential energy profile (in unit of kcal/mol) of the water-catalyzed proton transport between benzyl site and carbonate.

Besides, we have also computationally investigated the effect by increasing of the number of H₂O molecules and the results are listed in Figure 2 below. It shows that the binding energy and total barrier of the proton transfer increase along with increasing number of H₂O molecules, which tells that the H₂O molecule plays a role in hindering the reaction rather than stimulating it.

Figure 2. The binding energy, barrier energy of proton transfer and the total barrier energy respectively with different number N of H_2O molecules.

2. Would a series of mixtures be obtained when unsymmetrical substituted diaryliodonium salts and substituted alkenyl borate esters were examined in Figure 3? Could two different aryl radicals be detected if unsymmetrical substituted diaryliodonium salts were examined like Figure 4Ab?

Response: Thank you for your suggestions. It just afforded the single product **5ht**, (E)-1,3,5-trimethyl-2-(1-phenylprop-1-en-2-yl)benzene, when employing the unsymmetrical diaryliodonium salts **1h** ($ArI^+MesOTf^-$) and substituted alkenyl borate **2t**. And the phenyl radical was detected, as a contrast 2,4,6-trimethyl radical moiety was not detected.

Reviewer #2 (Remarks to the Author):

Functionalization of olefins via nonmetal catalysts remains a challenge in organic chemistry. In this work, a base-assisted transition-metal-free arylation method was reported by the author. This strategy enables vinyl pinacolboronates to be transformed in a one-pot procedure to give the desired diarylation olefins with diaryliodonium salts. Furthermore, a novel base-assisted radical-pair pathway was also proposed in this paper. Whilst the claim that the authors make is interesting to the community, I don't feel they've done enough to support and expand on it. A detailed reaction mechanism for the formation of carbonate-stabilized radical **7a** and followed radical addition steps

should also given by the DFT calculation as they proposed in figure 4B. I also don't distrust their energy results, however their discussion of the results could be improved. On the whole, I think this paper is suitable for publication on Communications Chemistry with minor revision.

The followings are several questions and suggestions

1. In Figure 2, the amount of base is different when K_2CO_3 and Li_2CO_3 were used respectively. Can author give a pellucid explanation on this phenomenon. May be the effect of Lewis acid K^+ or Li^+ should also be considered in the mechanism. Although the author have discussed the decomposition processes with K^+ in supporting information, the reaction model for this special step is not enough. The coordination number of K^+ is unsaturated and the single K^+ model should also be taken into account in the calculation (*J. Am. Chem. Soc.* **2017**, *139*, 29, 9909-9920).

Response: Thank you for the suggestions. Yes, the single K^+ model of $\text{Ph}_2\text{I}^+-\text{KCO}_3^-$ have now been taken into account in the calculation, where the estimation of the barrier is computationally investigated. The energy profile scan was added using $\text{I}-\text{C}_{\text{Ph}}$ as the reaction coordinates. This energy scan was also done for system of $\text{Ph}_2\text{I}^+-\text{CO}_3^{2-}$ and the results are shown in Figure 3 below. It reveals that when the $\text{I}-\text{C}_{\text{Ph}}$ bond reaches at 3.93\AA , the energy drops dramatically with the removing of phenyl radical and a generated unpaired electron on KCO_3^- anion transfers to I atom to get the KCO_3 radical. Finally, the KCO_3 radical and phenyl radical forms a radical pair by the cation- π interaction between potassium cation and benzene ring. The decomposition process requires an barrier of 51.8 kcal/mol , which is too high to occur under normal experimental condition. In contrast to $\text{Ph}_2\text{I}^+-\text{KCO}_3^-$, the barrier for $\text{Ph}_2\text{I}^+-\text{CO}_3^{2-}$ is 32.0 kcal/mol , which is relatively smaller indicating that this decomposition process is easier to take place. The structure of the product for the decomposition of $\text{Ph}_2\text{I}^+-\text{CO}_3^{2-}$ $\text{Ph}_2\text{I}^+-\text{KCO}_3^-$ are shown in Figure 4.

Figure 3. The potential energy profile (in unit of kcal/mol) of the decomposition of $\text{Ph}_2\text{I}^+-\text{CO}_3^{2-}$ and $\text{Ph}_2\text{I}^+-\text{KCO}_3^-$ in the gas phase.

The decomposition of $\text{Ph}_2\text{I}^+-\text{CO}_3^{2-}$ give the halogen bond complex radical pair $\text{PhI}-\text{CO}_3^{\cdot-}-\text{Ph}\cdot$, where the PhI and radical pair $\text{CO}_3^{\cdot-}-\text{Ph}\cdot$ bound by halogen bond, and of $\text{Ph}_2\text{I}^+-\text{KCO}_3^-$ give the radical pair $\text{KCO}_3^{\cdot-}-\text{Ph}\cdot$ bound to PhI also by the halogen bond. The I-O bond in the two decomposed product is 2.73Å and 2.97Å respectively. The phenyl trapped by TEMPO comes from the further step of $\text{PhI}-\text{CO}_3^{\cdot-}-\text{Ph}\cdot$ decomposition, where the phenyl radical bound to $\text{PhI}-\text{CO}_3^-$ by weak interaction between the hydrogen atom in phenyl radical and oxygen atoms in CO_3^{2-} moiety.

Figure 4. The structures of decomposed $\text{Ph}_2\text{I}^+-\text{CO}_3^{2-}$ and $\text{Ph}_2\text{I}^+-\text{KCO}_3^-$.

We have also considered the situation that reaction starts from the binding of KCO_3^- , the incomplete solvated product of K_2CO_3 in water, to vinylboronate **2a** and the results are shown in Figure 5. The binding of KCO_3^- to **2a** costs a small Gibbs free energy of 0.9kcal/mol. The addition of phenyl radical to $=\text{CH}_2$ group to get the intermediate III is calculated to be exothermic by 14.4kcal/mol. Then intermediate III reacts with $\text{PhI}-\text{CO}_3^-$ radical to produce the intermediate IV with PhI. The next step is the rate-determining one with barrier of 36.1kcal/mol. The Ha-O bond length in TS1 is 1.00Å and O-C bond is 1.53Å, which indicates that TS1 is a late transition state. The barrier height of the rate-determining step in the case of K^+ is slightly higher than the total barrier height according to the energy difference between TS2 and IV in the case without K^+ by about 3.2kcal/mol. The departure of $[\text{KCO}_3-\text{HCO}_3]^{2-}$ cluster from boron atom need to overcome a free energy barrier of 22.7 kcal/mol. This barrier is much larger than the case without K^+ . Therefore, from all these results, the favorable reaction pathway is the case without the participation of K^+ .

Figure 5. The energy profile (in unit of kcal/mol) of the reaction pathway level of B3LYP/def2TZVP(SMD)//B3LYP/def2SV(P)..

2. In figure 5, for the deprotonation step, the current transformation from intermediate IV seems unreasonable to give a relatively unstable intermediate V. Maybe the outer sphere deprotonation pathway is favorable than the current inner sphere pathway. Besides, another carbonate on the C atom can also be used for the deprotonation step. Furthermore, the detailed mechanism for the generation of radical 7a and aryl radical must be given in the PES.

Response: Thank you for your comments. According to the outer sphere deprotonation pathway, we select KCO_3^- and K_2CO_3 as exogenous bases respectively. Firstly, for the base of KCO_3^- , due to the static electrostatic repulsion between KCO_3^- and IV carrying -3 charge, although K^+ bonds to the carbonate which is connected to boron atom, the free energy of KCO_3^- -IV complex increases by 13.4 kcal/mol (Figure 6, right side). The barrier height of proton absorption is 31.2 kcal/mol. It requires in total 44.6 kcal/mol barrier height for the binding of KCO_3^- as shown in Figure 6. Thus the outer sphere deprotonation pathway can hardly go through in normal experimental condition. Secondly, when the exogenous base is K_2CO_3 , the two K^+ can bond to the two carbonate in intermediate IV and the complexation process is calculated to be exothermic by 11.4 kcal/mol (Figure 6, right side). Due to the weak basicity of K_2CO_3 , the barrier height of the proton abstraction is calculated to be as high as 38.9 kcal/mol which is not easy to occur either. Therefore, the outer sphere pathway is unlikely to take place for both of the two exogenous bases.

Figure 6. The potential energy profile (in unit of kcal/mol) of the process corresponding to the outer sphere deprotonation by KCO_3^- (left) and K_2CO_3 (right).

REVIEWERS' COMMENTS:

Reviewer #2 (Remarks to the Author):

The revised manuscript is suitable for publication on Communications Chemistry. while the picture in Figure 5 looks fuzzy. I suggest author to take as much space as you need to make your diagrams easy to understand.

Reviewer #2 (Remarks to the Author):

The revised manuscript is suitable for publication on Communications Chemistry. while the picture in Figure 5 looks fuzzy, I suggest author to take as much space as you need to make your diagrams easy to understand.

Response: Thank you for the helpful suggestion, according to which we have updated Figure 5 with more space and have added more detailed description in the illustration part as well.